# Field evidence of *Trypanosoma cruzi* infection, diverse host use and invasion of human dwellings by the Chagas disease vector in Florida, USA

Norman L. Beatty[1,2☯*], Chanakya R. Bhosale[3,4☯], Zoe S. White[3], Carson W. Torhorst[3], Kristen N. Wilson[3], Rayann Dorleans[3], Tanise M.S. Stenn[5], Keswick C. Killets[6], Rachel Curtis-Robles[6], Nathan Burkett-Cadena[5], Eva Nováková[7], Gabriel L. Hamer[8], Sarah A. Hamer[6], Samantha M. Wisely[2,3]

**1** Division of Infectious Diseases and Global Medicine, Department of Medicine, University of Florida College of Medicine, Gainesville, Florida, United States of America, **2** Emerging Pathogens Institute, University of Florida, Gainesville, Florida, United States of America, **3** Department of Wildlife Ecology and Conservation, University of Florida, Gainesville, Florida, United States of America, **4** Dr. Kiran C. Patel College of Osteopathic Medicine, Nova Southeastern University, Fort Lauderdale, Florida, United States of America, **5** Florida Medical Entomology Laboratory, Vero Beach, Florida, United States of America, **6** College of Veterinary Medicine & Biomedical Sciences, Texas A&M University, College Station, Texas, United States of America, **7** Faculty of Science, University of South Bohemia in Ceske Budejovice, Ceske Budejovice, Czech Republic, **8** Department of Entomology, Texas A&M University, College Station, Texas, United States of America

☯ These authors contributed equally to this work.
\* norman.beatty@medicine.ufl.edu

## Abstract

### Background

Triatomine bugs (Hemiptera, Reduviidae, Triatominae) are blood-sucking vectors of the protozoan parasite, *Trypanosoma cruzi,* which causes Chagas disease, a significant source of human morbidity and mortality in the Americas. Autochthonous transmission of Chagas disease in the United States is considered rare, despite evidence of *Triatoma* species harboring *T. cruzi*, invading homes, and biting occupants. In the southeastern United States, *Triatoma sanguisuga* is considered common, yet its distribution, host use, and *T. cruzi* infection are practically unknown in this region.

### Methodology/Principle findings

Using field sampling and community science programs from 2013 to 2023, we collected triatomines from peridomestic and domestic settings, identified them to species, analyzed for bloodmeals, and screened for *T. cruzi* infection and Discrete Typing Units (DTUs) TcI – TcVI utilizing molecular techniques. *Triatoma sanguisuga* (n = 310) were collected from 23 counties throughout the state, particularly in northern and central Florida. More than one third (34.6%) of *T. sanguisuga* were found inside a human dwelling, and 39.2% were collected by community members. *T. cruzi* infection

**Data availability statement:** All data are in the manuscript and/or Supporting information files.

**Funding:** NLB was supported by the University of Florida Research Opportunity Seed Fund (DRPD-ROSF2024), Mundo Sano Foundation (AWD08818), Khahn Dinh Fund for Chagas Disease Research (Fund 024569). RC-R was supported by the National Science Foundation (NSF) Graduate Research Fellowship Program (Grant No. 1252521). The Texas A&M University Kissing Bug Community Science program acknowledges funding from under cooperative agreement number UG4LM012345 with the University of North Texas Health Science Center - Gibson D. Lewis Library, and awarded by the DHHS, NIH, National Library of Medicine. GLH was supported in part by the Texas A&M AgriLife Research program. EN was supported in part by the Czech Science Foundation (Grant no. 21-10185M). The funders had no role in study design, data collection and analysis, decision to publish, or preparation of the manuscript.

**Competing interests:** The authors have declared that no competing interests exist.

was observed in 29.5% (88/298) of tested triatomines, with infection found in 12 of the 23 counties where triatomines had been collected. DTU-typing was successful for 47 of the *T. cruzi*-positive triatomines: 74.5% were infected with DTU TcI, 21.3% were infected with DTU TcIV, and 4.3% were co-infected with TcI and TcIV. Bloodmeal analysis of 144 *T. sanguisuga* found broad host use, including mammals (60%), ectothermic vertebrates (37%), and cockroaches (2.5%). Human blood meals contributed nearly a quarter (23%) of bloodmeals, indicating significant vector-human contact.

## Conclusion/Significance

Our field data from Florida demonstrate that *T. sanguisuga* is present near, and sometimes in, human dwellings, feeds upon humans, and is infected with multiple DTUs of *T. cruzi.* This indicates that the environment in the southeastern United States is suitable for autochthonous transmission of Chagas disease or that the human risk for *T. cruzi* infection is possible. The roles of ectotherms in *T. sanguisuga* and *T. cruzi* ecology also warrant further investigation.

## Author summary

Triatomines, which can spread Chagas disease to humans and other animals, are found throughout many U.S. states. Little is known about triatomines in Florida – our study aimed to learn more about where triatomines can be found in Florida, how often they are infected with the parasite that causes Chagas disease, and which animals triatomines feed on in Florida, including humans. From 2013-2023, utilizing our own field work and community science programs, we collected 310 triatomines from various regions in Florida. Both adult and immature triatomines were found, and about one-third of the triatomines were found inside human homes (35%). About 30% of the triatomines tested were infected with the Chagas disease parasite, *Trypanosoma cruzi*. Blood-fed triatomines mostly fed on mammals (60%, including humans), amphibians (27%), and reptiles (11%), but not on birds. Nearly a quarter of bugs (23%) had fed on human blood. Our research raises concerns for possible transmission of Chagas disease to humans from triatomines in Florida.

## Introduction

Triatomines (Hemiptera, Reduviidae, Triatominae), also known as kissing bugs, are found in at least 30 states across the lower half of the United States (U.S.) [1–4]. Triatomines are vectors of the parasite, *Trypanosoma cruzi*, which can cause Chagas disease in humans and other animals. Vector-borne transmission can occur via exposure to the insect's infected excreta (feces or urine) through a mucous membrane, a breach in the skin, or oral ingestion [5,6]. *Trypanosoma cruzi* infection primarily affects the cardiovascular system and can lead to chronic Chagas disease [7]. In the

U.S., vector-borne transmission of Chagas disease has been reported in at least eight states (California, Arizona, Texas, Missouri, Arkansas, Louisiana, Mississippi, Tennessee) [1,8,9] with *Triatoma sanguisuga* being implicated as the likely vector in at least two cases [10,11]. A recent epidemiological model assessing locally acquired infection has suggested that approximately 10,000 people are living with autochthonous Chagas, but a large-scale prevalence study has yet to be conducted in the U.S. [12].

Triatomine insects are hematophagous and feed either in association with a specific host or opportunistically in sylvatic conditions, yet some studies suggest over or under-utilization of different hosts based on their availability in peridomestic or domestic environments [13–15]. At least 11 species of triatomines have been documented in the U.S., including nine species of *Triatoma* and two species of *Paratriatoma* [1,2,16,17]. In the state of Florida, two distinct species (*Triatoma sanguisuga* and *Paratriatoma lecticularia)* are considered endemic [15–17]. One species, *Triatoma rubrofasciata*, was historically documented in the port city of Jacksonville, Florida. However, its current presence in the state remains uncertain [2,16].

*Triatoma sanguisuga* has been documented in 23 Midwest, Mid-Atlantic, and southern states. It has been collected as far north as Indiana and as far west as Wyoming [2–4]. This *Triatoma* species has the most expansive range of any triatomine species recorded within the U.S. and is known to invade human dwellings [2,18,19]. Early entomological studies from the 1940s and 1950s demonstrated that *T. sanguisuga* occurs throughout the state of Florida (32 of 67 counties), invades homes, and bites humans. Twenty years later (1969 – 1971) an extensive entomological survey of five counties (Alachua, Hardee Leon, Levy, and Orange) recovered nymphal and adult stages of *T. sanguisuga* in sylvatic environments, describing a close relationship with the Florida wood-roach (*Eurycotis floridana*) [20]. Nymphal stages of sylvatic Florida *T. sanguisuga* collected by Irons and Butler found these nymphs under loose bark of oak and pine trees, hollow stumps of trees or logs, and what they described as "brush piles" near the presence of "animal" runs [20]. The natural biomes of the sylvatic Florida *T. sanguisuga* has not been studied extensively and remains largely unknown in our region. The first record of *T. sanguisuga* harboring *Trypanosoma cruzi* in Florida was from a bug in Gainesville, Florida, collected from an outside door screen of a home in 1988 [21]. *Trypanosoma cruzi* was cultured from the insect and confirmed via detection of *T. cruzi* DNA using conventional PCR techniques. More recently, a community science program revealed *T. sanguisuga* infected with *T. cruzi* TcI and TcIV in Florida [14], and multiple species of mammals in Florida have been shown to be infected with *T. cruzi* in the region [22,23].

To our knowledge, no locally acquired human or domestic animal cases of Chagas disease have been reported in Florida. However, since all necessary components for transmission (triatomines, *T. cruzi*, and mammalian reservoirs) are present, potential risk of human and companion animal *T. cruzi* infection exists in Florida. To advance the understanding of the potential risk for autochthonous Chagas disease transmission, our multidisciplinary team investigated the ecology of triatomines in Florida. Using a combination of community submitted and investigator-initiated field collections, we explored the: 1) geographic distribution of triatomines in Florida, 2) occurrence of triatomine findings in human dwellings, 3) prevalence of *T. cruzi* among collected triatomines, and 4) bloodmeal sources of collected triatomines to determine host associations and potential routes of *T. cruzi* transmission.

## Methods

### Ethics statement

Community members submitted suspected triatomines voluntarily to the Texas A&M University Kissing Bug Community Science Program or to Dr. Norman Beatty at the University of Florida for identification and further study. Additional triatomines were collected at private residences after verbal permission, and a land use permit for triatomine collection at University of Florida Ordway Swisher Biological Station was provided (permit # OSR-20–12, OSR-21–12).

### Community science triatomine submissions

Community science programs developed at Texas A&M University (2013-present) [24] and the University of Florida (2020-present) [16,25] were used to solicit triatomines from the public. Community engagement strategies included

online educational websites on local triatomines, dedicated communication for inquires about triatomines via phone or email, online and in-person educational seminars on triatomines found in Florida and Chagas disease, local news outlet media and institutional social media promotion, awareness campaigns at local health fairs and events, and collaboration with non-profit organizations (https://kissingbugalliance.org) [16,24,25]. Physical triatomes which were placed in resign or pinned for display were brought to events to assist community members in recognizing the insect [25].Community members submitted insects for identification and provided some/all the following data: date, time of capture, geographic location (city, county), distance to human dwellings, location inside (i.e., kitchen, living room, etc.) or outside dwelling (i.e., porch, inside car, etc.), and potential human/domestic animal bite exposure.

Reports to the Texas A&M University program were submitted by community scientists, identified/confirmed, re-directed to state health department staff as appropriate, and otherwise processed as described previously [26]. For specimens brought to the University of Florida team from 2020-2023, both live and dead insect specimens were carefully removed from packages using sterile gloves and entomological forceps under a biosafety cabinet and placed in sterile tubes until identification. If live specimens were retrieved, they were stored within tubes in a freezer to kill them prior to identification. All packaging material was discarded in biohazard containers and properly discarded in case potentially infective triatomine excreta were present.

### Triatomine field collections

Field work by the University of Florida team from 2020-2023 included ultraviolet (UV) mercury vapor light trapping, night-time flashlight searching, and destructive sampling of decayed wood in peridomestic settings, or in sylvatic locations with suspected or known presence of triatomines (Fig 1). Most active manual search collections occurred in North-Central Florida, specifically around peridomestic areas at locations where triatomines had previously been submitted by community scientists. Ultraviolet light trapping was attempted on all nights and consisted of a portable 15-watt portable UV-A (longwave UV) light (wavelength between 350–400 nanometers) with AC and DC connector. When access to 120-volt power was available at a location we utilized an extension cord to the electricity source or

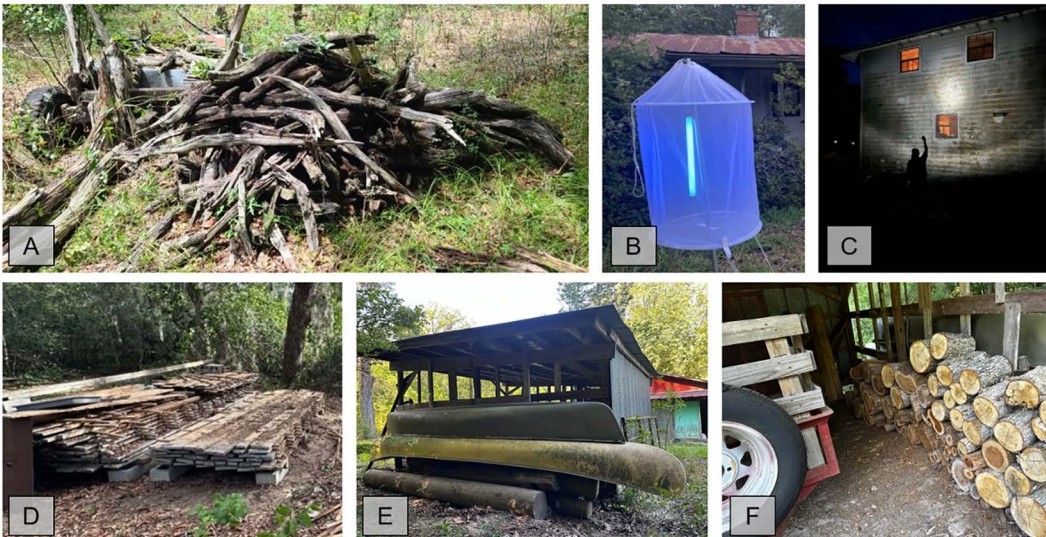

**Fig 1. Examples of locations where active manual searching for triatomines were attempted around human-dwellings in Florida.** (A) Natural cut woodpiles in yarda, (B) night UV sampling using UV light trap, (C) night-time lamp searching, (D) milled lumber woodpiles, (E) areas with potential animal-bloodmeal sources like underneath these outdoor canoes where rodents were harboring, (F) cut firewood in dry locations. (Photo Credits: Norman Beatty & Bernardo Peniche).

in more remote locations we connected to portable 12-volt sealed lead acid battery. UV light trapping involved two approaches each night with both set-ups. One UV light was suspended 4' above 6' x 6' white cotton sheet which was placed on flat on the ground. The other UV light trap was a portable fine mesh tent apparatus set-up above the ground approximately 4' with the UV light source inside the tent (Fig 1B). Active searching of the environment and human-made structures (wooden poles, vehicles, garage equipment, etc.), external walls around homes, cabins, garages, tree houses, dog kennels, and/or porches) using battery-powered lights were conducted for 4 hours beginning at sunset. Destructive sampling was conducted in the daytime in the environment by turning over, excavating, or sorting through wood piles (natural cut logs, stacked lumber, etc.), rocks, debris, or leaf litter near or over suspected host animal dens, burrows, or nests. Chicken coops, outside domestic chicken roosts, or suspected rodent nests were also examined thoroughly for triatomines. All triatomine specimens (live or dead) were placed in 100% ethanol.

**Triatomine processing**

Triatomine specimens sent to Texas A&M University were processed as previously described [26]. Life stage (adult or nymph) and sex of adults was noted. Nymphal instar was not determined. Bloodmeals were scored as 1–5 as previously described [26]. For the subset of specimens (n = 12) that were found in sleeping areas or that had possibly bitten a person, submitters were referred to the state health department; in some of those cases, although the specimen was known to be a *T. sanguisuga* (based on submitter's past experience with triatomines), the life stage was not noted for individual specimens (n = 3).

Triatomine specimens sent to the University of Florida were morphologically identified using a Leica light microscope (Leica, Buffalo Grove, IL, USA) and the most widely accepted Triatominae morphological key [15]. Life stage (1–5 nymphal instar, and adult) was noted, and sex of adult specimens were recorded. After identification, each bug was stored at -20°C or -80°C prior to further processing. Specimens were individually dissected in sterile petri dishes within a class-II biological safety cabinet. The surface of each specimen was rinsed 3–4 times with sterile 1M PBS solution, 3–4 times with sterile $H_2O$, surface-sterilized using 75% ethanol to clear any environmental debris or contamination and then let air dry. Using a flame sterilized scalpel and forceps, wings (if present), legs, head, and thorax of the insect were carefully removed and then stored at -80°C. If a specimen was engorged or had evidence of bloodmeal, using sterile forceps, the full abdomen was first delicately squeezed in a separate 1.5ul sterile centrifuge tube to retrieve any liquid abdomen contents or bloodmeal. The presence, description (liquid, clear, dry, chunks), and color (red, brown, black, yellow) of bloodmeal was recorded for future bloodmeal analysis. The abdomen was then cut medially using the scalpel and one half was placed in a separate 1.5ul sterile centrifuge tube for storage in -80°C. Contents of the other half abdomen were delicately removed and placed in the same tube as any liquid fecal contents. DNA of combined blood, hind-gut, and fecal matter (if present) was extracted used the Qiagen Gentra Pure Gene kit (Qiagen, Hilden, Germany). Total genomic DNA was extracted from the sample using InstaGene Matrix (Product #: 7326030; Bio-Rad Laboratories Inc.: Hercules, CA, U.S.) following the manufacturer protocol. If high red blood cell content was present, the contents were first placed in a 750ul RBC cell lysis solution for 10 minutes to avoid heme contamination. Contents were then placed in a solution of 600ul cell lysis and 15 µl Proteinase K (Prot K) (20 mg/ml) and incubated at 56°C on a shaker incubator (400RPM) for 48 hours. If large chunks of contents were still present after 48 hours, more additional Prot K was added, and samples were incubated an additional 24 hours at 56°C. Total DNA was extracted, and the pelleted DNA was resuspended in 50–150ul of AE buffer depending on pellet size. Using the Nanodrop One (Thermo Fisher Scientific, Carlsbad, California, U.S.), DNA quality and quantity were calculated for each sample, and samples were diluted to a working concentration of 20ng/ul. All DNA samples were stored at -20°C until polymerase chain reaction (PCR) was performed and then at -80°C for long term storage.

## Trypanosoma cruzi molecular detection and DTU identification

DNA samples from triatomine specimens submitted to Texas A&M University were screened for *T. cruzi* infection using Cruzi1/2 primers and Cruzi3 probe on a real-time qPCR assay [27,28], similar to previous work [14]; samples with cycle threshold (Ct) values of < 33 were considered positive, and samples with Ct values of ≥ 33 were considered negative. *T. cruzi* DTUs were identified using a real-time SL-IR targeting qPCR [29].

DNA samples from triatomine specimens submitted to the University of Florida team were screened for *T. cruzi* and underwent further DTU classification which were adapted from previous studies among triatomines, humans, domestic dogs, and wildlife [22,23,27–30]. For each reaction, 2ul of the working 20ng/ul sample, along with 18ul of master mix was used with an exogenous amplification control (10x Exo IPC Mix, 50 IPC Exo) (Thermofisher Scientific, Waltham, MA, U.S.) to identify possible amplification inhibitors of triatomine DNA with primers and probe retrieved from two prior studies [22,28]. Each assay contained negative and positive controls. Negative control consisted of molecular grade water, and an exogenous amplification negative with molecular grade water as a template control. Three *T. cruzi* positive controls were also run in a calculated dilution series of extracted DNA from cell culture of *T. cruzi* TcI (Dm28c) [23,24,30]. A sample was considered positive for *T. cruzi* if the Ct value was ≤ 38, or suspect postive if Ct value from 38-40. All suspect positives and those samples that had no amplification in the exogenous amplification control were rerun and rescreened using 1:5 and 1:10 molecular water dilutions to remove any possible errors or amplification inhibitors from the first run [23,24,30]. If subsequent reruns amplified with Ct value of ≤38 they were labeled as positive for *T. cruzi* infection, but if the suspect positive sample reamplified at a Ct value ≥38, then the sample was labeled as negative. For epidemiological importance, a Wilson 95% CI was calculated using EPI tools (https://epitools.ausvet.com.au/ciproportion) to calculate confidence limits for total sample proportion calculations of *T. cruzi* presence in triatomines.

For specimens tested at the University of Florida, all positive *T. cruzi* samples were run through a multiplex stepwise qPCR to identify DTUs TcI – TcVI [28]. For our DTU identification assays, we modified the first multiplex SL-IR assay which classifies DTU TcI, to a duplex, while all other qPCRs were kept from their original description [29]. Modified fluorescent probe information used for our assays has also been published [22,23]. Each DTU assay utilized a negative template control using molecular H$_2$O. Positive controls using a TcI (Dm28c strain) from *T. cruzi* cell culture isolate were used for TcI identification, and synthetically constructed plasmids for TcII – TcVI were used for all other assays. DTU identification was attempted three times on samples; including the 20ng/ul working aliquot and using the 1:5 and 1:10 dilution series, respectively. After these steps were attempted, if DTU identification for a sample was unsuccessful, it was designated as untypable. Co-infections of DTUs were also noted if retrieved in our assays. An ABI 7500 Fast Real-Time PCR System Machine (Thermofisher Scientific, Waltham, MA, U.S.) was used for all *T. cruzi* screening and DTU identification qPCRs.

## Triatomine bloodmeal analysis

Blood-engorged triatomines (n = 156/298) collected by the University of Florida and Texas A&M teams were analyzed individually using polymerase chain reaction (PCR) and Sanger sequencing to determine vertebrate host species. Each PCR reaction consisted of 12.5 μL Platinum Green 2X Master Mix (Invitrogen: Carlsbad, CA, U.S.), 9 μL molecular-grade water, 0.5 μL forward primer (20 μM), 0.5 μL reverse primer (20 μM), and 2.5 μL template for a total volume of 25 μL. Four primer sets were used, in sequence (Table 1). Since mammals are considered important hosts for triatomines in Florida, a primer set targeting mammalian 16S rRNA gene was first used [31]. This primer set also successfully amplifies amphibian DNA [32,33]. Samples that did not amplify with the first primer set were then run using primer sets that target 16S rRNA of reptiles [34,35] followed by two primer sets targeting Cytochrome b gene of birds [36]. PCR thermocycling conditions for bloodmeal analysis were identical to previously published conditions [32,37,38]. PCR products were run on 1% agarose gel, and amplicons were sequenced using Sanger method by Eurofins Genomics (Ebersberg, Germany). Resulting sequences were compared to available sequences in NCBI GenBank [39] using the Basic Local Alignment Search Tool (BLASTn) (http://www.ncbi.nlmn.nih.gov/BLAST).The minimum sequence similarity for species-level

**Table 1. Polymerase chain reaction primers used in host bloodmeal identification.**

| Primer pair | Sequence | Target taxa | Amplicon size (bp) |
|---|---|---|---|
| L2513<br>H2714 | 5'-GCCTGTTTACCAAAAACATCAC-3'<br>5'-CTCCATAGGGTCTTCTCGTCTT-3' | Mammals, amphibians | 210 |
| L0<br>H1 | 5'-GGACAAATATCATTCTGAGG-3'<br>5'-GGGTGGAATGGGATTTTGTC-3' | Birds | 220 |
| L0<br>H0 | 5'-GGACAAATATCATTCTGAGG-3'<br>5'-GGGTGTTCTACTGGTTGGCTTCC-3' | Birds | 589 |
| 16L1<br>H3056 | 5'-CTGACCGTGCAAAGGTAGCGTAATCACT-3'<br>5'-CTCCGGTCTGAACTCAGATCACGTAGG-3' | Reptiles | 450 |

identification and analysis for this manuscript was 95%, although average similarity to the closet match was high (98.3%). Matches less than 95% were still noted in data but not included in bloodmeal analysis results. The chi-square test of independence and Fisher's exact test were used to analyze differences in triatomines captured indoors and outdoors among bloodmeal hosts [32,38].

## Results

### Triatomine collections

A total of 310 triatomines were collected from 38 locations throughout 23 Florida counties; four additional locations in southern Florida were searched without finding triatomines (Fig 2). All specimens were morphologically identified as *T. sanguisuga* (Fig 3). A total of 152 adults (72 females, 73 males, 7 undetermined sex), 154 nymphs (4 first instars, 19 second instars, 34 third instars, 30 fourth instars, 65 fifth instars, 2 undetermined instar), and 4 unidentified of any life stage. Eggs were also found in adult females during the dissection process. Across all sampling locations, the majority of triatomines (64.7%) were captured outside a human dwelling, while 34.6% were found inside human dwellings or structures, including one bug located inside an automobile and one bug located inside a portable metal storage container (Table 2). Overall, all successful bug collections and submissions in this study occurred inside, on, or closely outside human dwellings or human-made structures. No triatomines were collected in purely sylvatic locations, including South Florida.

Community science submissions accounted for 39% (122/310) of the collected triatomines, submitted by 31 individual contributors. The majority were adult insects, with a small proportion of nymphs (n = 5; 4.1%). Most community science submissions occurred from April to August (93%), with a peak in June (Fig 4). The number of bugs retrieved from community science submissions ranged from 1-51 per submission per household and were found in various areas inside (i.e., bedroom, living room, kitchen) or outside (walls/porches) of human dwellings (Table 4). Although not retrieved from every household where triatomines were collected, overall, twelve households among the 31 submitters confirmed the presence of companion animals (i.e., cats or dogs) at the home or other domestic livestock near the home or on their property. Five submissions recorded human or canine bite exposures by triatomines.

Host identity was determined through PCR-based assays targeting 16S and Cytochrome b genes followed by Sanger sequencing of amplicons. Indoor refers to triatomine found inside human dwelling and outdoor would be outside the human dwelling. The percentage indoor and outdoor (nymphs and adults) refers the number (n) found in each location with that bloodmeal detected. Total (n) percentage refers to that bloodmeal among all tested triatomines. The percentage match corresponds to bloodmeal amplicon sequence average similarity to reference sequences in GenBank.

Daylight destructive sampling accounted for a total of 25 encounters for approximately 138 hours. Night-time collecting, which included both searching and UV light trapping, accounted for a total of 12 encounters constituting 48 hours of searching. Both daylight destructive sampling and night-time collecting mainly occurred in north and central Florida regions around areas where triatomines had been collected by community scientists. Active manual searching yielded the greatest number of

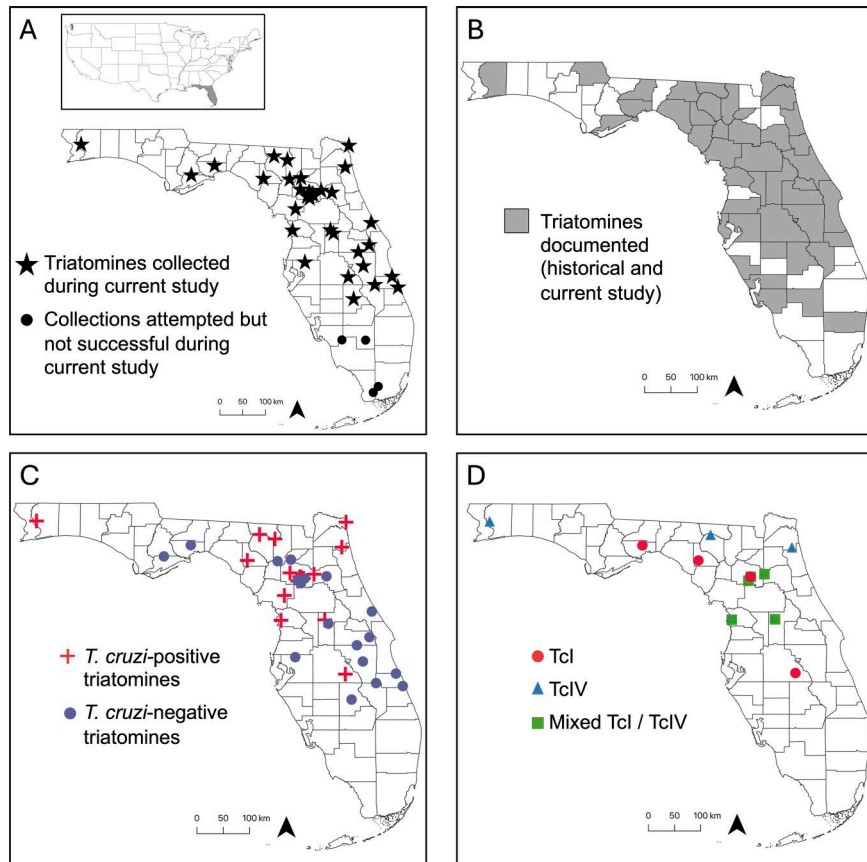

**Fig 2. Triatomine collections and *T. cruzi* testing results from Florida, U.S.A.** (A) Locations of triatomine collections and collection attempts during current study. (B) Locations where triatomines have been recorded historically [16] and during current study. The current study added 9 counties to this distribution map: Franklin, Hamilton, Lake, Nassau, Osceola, Polk, Santa Rosa, Union, Wakulla. (C) *T. cruzi* testing results for triatomines collected during current study. (D) *T. cruzi* DTU results from triatomines collected during this study. (Mixed TcI/TcIV refers to locations where both DTUs were detected but not necessarily from the same triatomine). Maps were created in QGIS (version 3.28) and base map data from the U.S. Census Bureau [40,41].

triatomines (both adults and nymphs), with 51.6% (160/310) and 9.0% (28/310) respectively, and were successfully attempted at 14/20 locations (Table 3) where we were permitted for this sampling. Unsuccessful collection attempts (10 separate days; 2021–2023; approximately 40 hours) occurred in several areas of south Florida where suspected habitats may be present (Fig 2). Most successful destructive sampling occurred within woodpiles of natural oak/cedar logs over suspected rodent/animal dens, or in covered lumber or firewood. Successful active manual searching yielded mainly nymphs, including triatomine exuviae via woodpiles, with some areas where adult triatomines were also collected via searching and hand-collecting in porches, around human dwellings, garages, or in certain woodpiles (Table 4). The closest successful capture of live nymphs outside a home was in a covered oak wood pile five meters from a front porch. The closest successful hand-collection of an adult female triatomines occurred underneath dog bedding located in a netted-back porch within two meters to a primary entrance into the home. The abdomen of a suspected nymph was also found in a broken windowsill in a location where dispersing adult triatomines were previously found via night-collecting. At all active manual searching locations or human dwellings, there were about 1–8 different destructive sampling or searching areas/colonies (i.e.,. woodpiles, garages, back-porch) attempted, and successful triatomine colonies yielded 1–55 dead or alive bugs per destructive sampling session. One colony of triatomines

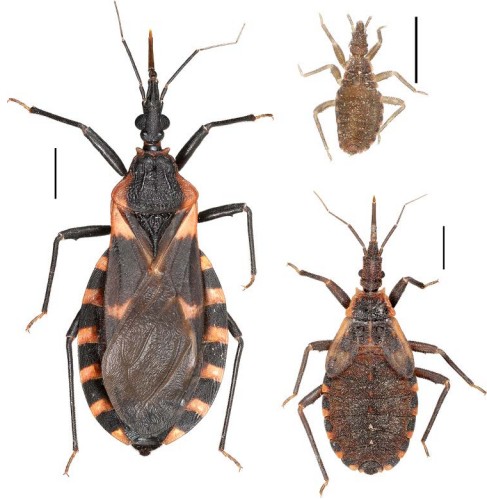

**Fig 3. Florida *Triatoma sanguisuga* adult female (left) with 1st instar (top right) and 5th instar nymphal stages collected at a Florida home.** (Scale bar is 2mm). (Photo Credit: Nathan Burkett-Cadena).

**Table 2. Triatomine collection type or environment location of capture.**

| | Total triatomines collected (n = 310; % total) | Triatomine life stage | *T. cruzi* infected triatomines (n = 88; % total) |
|---|---|---|---|
| **Type of Capture** | | | |
| Community science | 122 (39.3%) | Nymphs and adults | Yes (26.1%) |
| Night hand collecting | 28 (9.1%) | Adults only | Yes (18.2%) |
| Destructive sampling | 160 (51.8%) | Nymphs and adults | Yes (55.7%) |
| **Location of Capture** | | | |
| Inside human dwelling | 108 (34.6%) | Nymphs and adults | Yes (23.9%) |
| Outside human dwelling | 200 (64.7%) | Nymphs and adults | Yes (76.1%) |
| Inside car or storage container | 2 (0.65%) | Adults only | Not tested |

was discovered within 30 meters of a human dwelling within a covered, milled oak lumber pile, with all instars (1–5) of *T. sanguisuga,* including an adult female, and triatomine exuviae found at this site.

Compared to destructive sampling, night-time collecting yielded mainly dead, or alive adult-flying/scurrying triatomines found on the outside walls, screened porches, or inside screened patios of human dwellings or on wooden human-made structures using lamp-searching in a specified sampling location (Tables 2 and 3). Almost all night-time collections of live adults occurred around dusk until 10pm. Although ultraviolet light trapping was able to attract other night-time flying arthropods, no triatomines were found directly on the UV light tent trap or on white sheet during night-time collections at attempted locations.

### Triatomine bloodmeal analysis

Overall, bloodmeal analyses revealed twelve species of mammals, seven species of amphibians, and five species of reptiles (Table 4). A total of 92.3% (n = 144/156) of the bloodmeals were identified with >95% confidence in species-levels detection (n = 12 bloodmeals <95% (i.e., n = 1 with 89.71% match to *Sus scrofa* [wild pig], n = 1 with 88.1% match to *Odocoileus virginianus [white-tailed deer])*. Florida *T. sanguisuga* collected from indoors and outdoors were found to feed upon a diverse array of vertebrates and one invertebrate (Fig 5). Most bloodmeals (n = 144) were from mammals (59.7%), followed by amphibians (27.1%), and reptiles (11.1%). Three meals analyzed from triatomines revealed

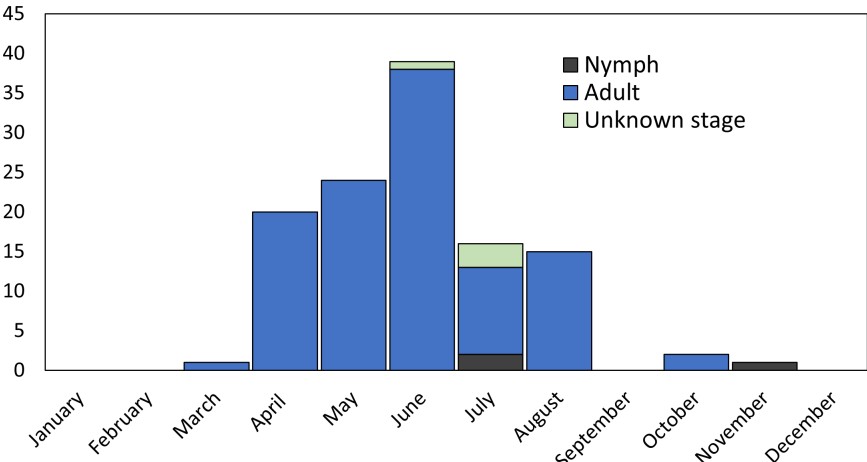

**Fig 4. Community science collected Florida triatomines (n = 122) found inside or outside a human dwelling by month of collection.** (Adult, n = 111; Nymph; n = 3; Unknown stage, n = 4; Unknown month, n = 4).

**Table 3. Specified locations where triatomines were captured by community scientists or active manual searching by research team at human dwellings in Florida. (If no specific area was given it was referred to as unspecified).**

| | | *T. cruzi* negative triatomine | *T. cruzi* positive triatomine |
|---|---|---|---|
| **Indoor** | Living Room | Inside bug zapper; within spider web, crawling on wall. | In lights on wall; Unspecified*. |
| | Bedroom | Crawling on bed; under pillow; inside pillow covering; Crawling on curtain; Children bedroom; Unspecified. | Bunk house bedroom windowsill; On bed, found engorged on cabin floor in bedroom. |
| | Kitchen | | Inside kitchen sink; Crawling on kitchen counter; Crawling on kitchen curtain; Inside an old metal can found in kitchen; Unspecified. |
| | Bathroom | Crawling in bathroom; Bathroom sink; Under bathmat near shower. | Unspecified. |
| | Other indoor areas | Inside washing machine; Apartment ceiling near foyer; crawling near front door of house; Brought into house on drying beach towels located on balcony; Wall of house inside closed porched area. | In dog food bowl inside house; Stairs to apartment; Unspecified. |
| **Outdoor** | Other structures | Outdoor open barns storing agriculture equipment; Outdoor garage structure; Inside automobile located in nature park; Inside metal storage container on nature preserve. | Wooden tree house where humans had slept; Open wooden structure for lodging at "fish camp" |
| | Connected to/ near dwellings | Open front porch; Second floor open porch crawling toward sitting homeowner; Inside screened patio wall bunkhouse in nature preserve; Triatomine was collected after it flew on collector's body at night; Resting on outside wall of home. | Open back porch near wood line; On wooden fire house lookout; Inside dog kennel on open back porch; On outside patio screened netting connected to home; In spider web-on outside front stairs of home; Resting on wooden pillar of outdoor car garage; Windowsill of garage. |
| | Wood piles near home | Lumber milled oak and cypress wood plank pile inside covered outdoor garage. | Covered oak woodpile (logs) near home; Covered natural oak firewood pile (split) in open shed; Lumber milled oak plank pile over suspected animal den(s); Natural oak/cedar woodpiles with animal sightings by homeowner; Utility pole wood piles near open outdoor canoe shed and storage. |

*If no specific area was given it was referred to as unspecified.

**Table 4.** *Triatoma sanguisuga* bloodmeal hosts from Florida, United States.

| Host class | Host common name | Adult Indoor | | Adult Outdoor | | Nymph Outdoor | | Total | | Match % |
|---|---|---|---|---|---|---|---|---|---|---|
| | | n | % | n | % | n | % | n | % | % |
| **Amphibia** | | | | | | | | | | |
| | *Anaxyrus terrestris/fowleri* (Southern toad/Fowler's toad) | 0 | 0 | 4 | 28.6 | 18 | 20.0 | 22 | 15.3 | 97.9 |
| | *Gastrophryne carolinensis* (Eastern narrow-mouthed toad_ | 0 | 0 | 0 | 0.0 | 5 | 5.6 | 5 | 3.5 | 98.2 |
| | *Dryophytes cinereus* (Green treefrog) | 4 | 10 | 0 | 0.0 | 0 | 0.0 | 4 | 2.8 | 97.7 |
| | *Osteopilus septentrionalis* (Cuban treefrog) | 2 | 5 | 0 | 0.0 | 0 | 0.0 | 2 | 1.4 | 98.0 |
| | *Eleutherodactylus planirostris* (Greenhouse frog) | 1 | 2.5 | 0 | 0.0 | 1 | 1.1 | 2 | 1.4 | 97.0 |
| | *Dryophytes femoralis* (Pine woods treefrog) | 2 | 5 | 0 | 0.0 | 0 | 0.0 | 2 | 1.4 | 96.5 |
| | *Hyla squirella* (Squirrel treefrog) | 1 | 2.5 | 1 | 7.1 | 0 | 0.0 | 2 | 1.4 | 97.0 |
| | *Total Amphibia* | 10 | 25 | 5 | 35.7 | 24 | 26.7 | 39 | 27.1 | NA |
| **Mammalia** | | | | | | | | | | |
| | *Homo sapiens* (Human) | 18 | 45 | 2 | 14.3 | 12 | 13.3 | 32 | 22.2 | 98.0 |
| | *Didelphis virginiana* (Virginia opossum) | 1 | 2.5 | 0 | 0.0 | 27 | 30.0 | 28 | 19.4 | 99.2 |
| | *Dasypus novemcinctus* (Nine-banded armadillo) | 0 | 0 | 2 | 14.3 | 5 | 5.6 | 7 | 4.9 | 98.8 |
| | *Procyon lotor* (Northern raccoon) | 5 | 12.5 | 0 | 0.0 | 0 | 0.0 | 5 | 3.5 | 98.5 |
| | *Sylvilagus floridanus* (Eastern cottontail rabbit) | 1 | 2.5 | 2 | 14.3 | 1 | 1.1 | 4 | 2.8 | 99.6 |
| | *Canis latrans* (Coyote) | 2 | 5 | 0 | 0.0 | 0 | 0.0 | 2 | 1.4 | 96.3 |
| | *Canis lupus familiaris* (Domestic dog) | 1 | 2.5 | 1 | 7.1 | 0 | 0.0 | 2 | 1.4 | 96.7 |
| | *Neotoma floridana* (Eastern woodrat) | 0 | 0 | 0 | 0.0 | 2 | 2.2 | 2 | 1.4 | 96.3 |
| | *Peromyscus gossypinus* (Cotton mouse) | 0 | 0 | 0 | 0.0 | 1 | 1.1 | 1 | 0.7 | 98.6 |
| | *Sciurus carolinensis* (Eastern gray squirrel) | 0 | 0 | 1 | 7.1 | 0 | 0.0 | 1 | 0.7 | 96.0 |
| | *Felis catus* (House cat) | 1 | 2.5 | 0 | 0.0 | 0 | 0.0 | 1 | 0.7 | 97.0 |
| | Human + smokybrown cockroach | 0 | 0 | 0 | 0.0 | 1 | 1.1 | 1 | 0.7 | 98.6; 100 |
| | *Total Mammalia* | 29 | 72.5 | 8 | 57.1 | 49 | 54.4 | 86 | 59.7 | NA |
| **Reptilia** | | | | | | | | | | |
| | *Plestiodon laticeps* (Broad-headed skink) | 0 | 0 | 2 | 14.3 | 10 | 11.1 | 12 | 8.3 | 98.8 |
| | *Nerodia fasciata* (Banded watersnake | 0 | 0 | 0 | 0.0 | 1 | 1.1 | 1 | 0.7 | 99.8 |
| | *Plestiodon inexpectatus* (Southeastern five-lined skink) | 0 | 0 | 0 | 0.0 | 1 | 1.1 | 1 | 0.7 | 99.1 |
| | *Hemidactylus mabouia* (Tropical house gecko) | 1 | 2.5 | 0 | 0.0 | 0 | 0.0 | 1 | 0.7 | 98.9 |
| | *Coluber constrictor* (Eastern racer) | 0 | 0 | 0 | 0.0 | 1 | 1.1 | 1 | 0.7 | 96.6 |

*(Continued)*

**Table 4.** (Continued)

| | | Adult | | | | Nymph | | | | Match |
| | | Indoor | | Outdoor | | Outdoor | | Total | | % |
| Host class | Host common name | n | % | n | % | n | % | n | % | |
| | *Total Reptilia* | 1 | 2.5 | 2 | 14.3 | 13 | 14.4 | 16 | 11.1 | NA |
| **Insecta** | | | | | | | | | | |
| | *Periplaneta fuliginosa* (Smokybrown cockroach) | 0 | 0 | 0 | 0.0 | 3 | 3.3 | 3 | 2.1 | 98.9 |
| | *Total Insecta* | 0 | 0 | 0 | 0.0 | 3 | 3.3 | 3 | 2.1 | NA |

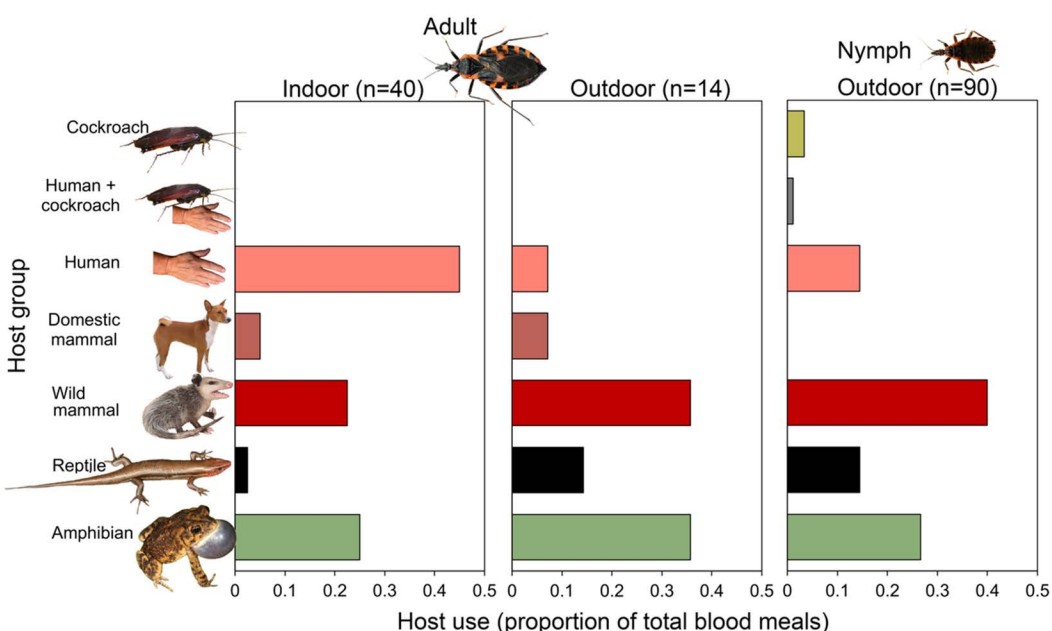

**Fig 5. Host associations of *Triatoma sanguisuga* collected from Florida.** PCR-bloodmeal analysis targeting vertebrate mitochondrial and ribosomal genes of DNA extracted from triatomine digestive tracts. Triatomines were collected from the peridomestic environment across 23 counties (2013-2023) in Florida, U.S.

smokybrown cockroach (*Periplaneta fuliginosa*) DNA, including one bloodmeal which contained both human and cockroach DNA. The smokybrown cockroach DNA was incidentally and unexpectedly amplified using primers that target 16S rRNA of reptiles [33,34]. Further sanger sequencing of these amplicons revealed and overall, 98.9% match of the smokybrown cockroach when compared to available sequences in the NCBI GenBank for *Periplaneta fuliginosa*. The most fed upon individual host species were humans (22.2%), Virginia opossum (*Didelphis virginiana* [19.4%]), southern/ Fowler's toad (*Anaxyrus terrestris/ fowleri* [15.3%]), and broad-headed skink (*Plestiodon laticeps* [8.3%]) (Table 4). Humans constituted a larger fraction of bloodmeals from adult triatomines captured indoors (45.0%) than adult triatomines (14.3%) or nymphal stages (13.3%) captured outdoors (Fig 5). Conversely, wild mammals contributed larger fractions of bloodmeals of adult triatomines (35.7%) and nymphal stages (40.0%) captured outdoors than adult triatomines captured indoors (22.5%). Reptiles also contributed to larger fractions of bloodmeals of adult triatomines (14.3%) and nymphal stages (14.4%) captured outdoors as opposed to adult triatomines captured indoors (2.5%). Amphibians contributed a relatively large fraction of bloodmeals for adult and nymphal stages, both indoors (25.0% of adults) and outdoors (35.7% of adults

and 26.7% of nymphs). Surprisingly, dogs contributed a relatively low percentage of bloodmeals of adults (2.5-7.1%) or nymphal stages (0.0%).

Significant differences in bloodmeal distribution were observed between triatomines captured indoors and outdoors. Amphibians were significantly more common in outdoor bloodmeals ($\chi^2 = 36.49$, d.f. = 7, $P < 0.001$), as were mammals ($\chi^2 = 42.32$, d.f. = 11, $P < 0.001$) and reptiles ($\chi^2 = 16.0$, d.f. = 4, $P = 0.003$). For amphibians, toads (*Anaxyrus terrestris/ Anaxyrus fowleri)* contributed 68.8% of amphibian bloodmeals, but were not found among bloodmeals from the 40 tested triatomines captured indoors (0%). Among mammals, Virginia opossum was a major host for triatomines collected outdoors and constituted nearly half (47.4%) of all mammalian bloodmeals. However, among triatomines captured indoors the Virginia opossum contributed just 3.4% of total mammal bloodmeals. Broad-headed skink contributed 68.8% of reptile bloodmeals but were not found among bloodmeals from triatomines captured indoors (0%). Humans constituted a significantly larger fraction of bloodmeals from adult *T. sanguisuga* captured indoors (42.3%) compared to outdoors (7.1%), according to Fisher's exact test (p = 0.024). Adult *T. sanguisuga* took a significantly larger fraction of total bloodmeals from humans (34.9%) than did nymphal stages (14.4%, Fisher's exact test, p = 0.003).

## Trypanosoma cruzi infection and DTU analysis

In this study, n = 298/310 *T. sanguisuga* were tested for *T. cruzi* infection via qPCR and subsequent molecular DTU analysis. In total, n = 88/298 (29.53%) of these triatomines (of all life stages, sex, and instars) across 12/23 counties were *T. cruzi*-positive (Fig 2C and Table 5). *T. cruzi* was detected in at least one triatomine from each reported location. Of the 88 *T. cruzi* positive triatomines, over half (n = 47 or 53.41%), were typeable for a *T. cruzi* DTU, with typable specimens (both nymphs and adults) infected with TcI (74.47%), TcIV (21.28%), or TcI + IV coinfection (4.26%) (Fig 2D and Table 6). Overall, *T. cruzi* positive triatomines (both adults and nymphs) were found using both community science and active manual searching in multiple locations inside or outside human dwellings and close to human and companion animal activity (Tables 2 and 3). Our bloodmeal analysis showed that *T. cruzi*-positive triatomines fed on humans, companion animals, and wildlife reservoirs (Table 7). A total of 27% (n = 39/144) of bloodmeals correlated with detection of *T. cruzi* in the same triatomine. It revealed that *Didelphis virginiana* (Virginia opossum) was the most common host with which we simultaneously detected *T. cruzi* (n = 12/39). Interestingly, bloodmeals with southern toad (Fowler's Toad) were also found with *T. cruzi* detection in 20.5% (n = 8/39) of these bloodmeals. The nine-banded armadillo was only associated with *T. cruzi* DTU TcI (n = 6/39). Among the *T. cruzi* which were untypable (n = 17/39) we found simultaneous bloodmeal host detection present with the Virginia opossum (n = 8/17), broad-headed skink (n = 2/17), human (n = 2/17), southern toad (n = 2/17), squirrel treefrog (n = 1/17), house cat (n = 1/17) and the eastern woodrat (n = 1/17). *T. cruzi* DTU TcIV was detected in two triatomines with a bloodmeal (broad-headed skink and squirrel treefrog). One bloodmeal which detected only human was also associated with the detection of a mixed infection with *T. cruzi* DTU TcI and TcIV among that triatomine.

**Table 5. Infection percentage of triatomines tested (n = 298) for *T. cruzi* infection and *T. cruzi* DTU.**

| Tested Type (*T. cruzi* or *T. cruzi* DTU) | # of triatomines positive/ # of triatomines tested or typable for DTU (%) | 95% Wilson CI |
|---|---|---|
| *Trypanosoma cruzi* DNA | 88/298 (29.53%) | [24.64%-34.94%] |
| *DTU Typeable | 47/88 (53.41%) | [43.06%-63.47%] |
| -TcI DTU Only | 35/47 (74.47%) | [60.49%-84.75%] |
| -TcIV DTU Only | 10/47 (21.28%) | [11.99%-34.90%] |
| -TcI TcIV DTU Coinfection | 2/47 (4.26%) | [1.17%-14.25%] |

*Note that 41/88 (46.59%) of *T. cruzi* positive bugs were DTU untypable.

Wilson 95% confidence interval is given for all percentages.

**Table 6. Number of Florida *T. sanguisuga* adults (male vs female) vs nymphs (by instar) testing positive *T. cruzi* and further detection of specific *T. cruzi* DTU.**

| Triatomine life stage | Total specimens collected | Total specimens tested for *T. cruzi* | *T. cruzi* infected of tested bugs | *T. cruzi* TcI | *T. cruzi* TcIV | Mixed *T. cruzi* TcI & TcIV | *T. cruzi* untypable DTU |
|---|---|---|---|---|---|---|---|
| Adult | 152 | 141 | 20.4% (40/141) | 32.5% (13/40) | 25.0% (10/40), | 2.5% (51/40) | 40.0% (16/40) |
| Male | 73 | 68 | 27.9% (19/68) | 31.6% (6/19) | 15.8%(3/19) | 5.3% (1/19) | 47.4% (9/19) |
| Female | 72 | 68 | 26.5% (18/68) | 27.8% (5/18) | 38.8% (7/18) | ND | 33.3% (6/18) |
| Sex unidentified | 7 | 5 | 60.0% (3/5) | 66.7% (2/3) | ND | ND | 33.3% (1/3) |
| Nymphs | 154 | 154 | 31.2% (48/154) | 45.83% (22/48) | ND | 2.0% (1/48) | 52.0% (25/48) |
| 1st instar | 4 | 4 | 25.0% (1/4) | ND | ND | ND | 100.0% (1/1) |
| 2nd instar | 19 | 19 | 21.0% (4/19) | 25.0% (1/4) | ND | ND | 75.0% (3/4) |
| 3rd instar | 34 | 34 | 32.4% (11/34) | 9.0% (1/11) | ND | 9.0% (1/11) | 81.8% (9/11) |
| 4th instar | 30 | 30 | 33.3% (10/30) | 40.0% (4/10) | ND | ND | 60.0% (6/10) |
| 5th instar | 65 | 60 | 35% (21/60) | 76.1% (16/21) | ND | ND | 23.8% (5/21) |
| Instar unidentified | 2 | 2 | 50.0% (1/2) | ND | ND | ND | 100.0% (1/1) |
| Life stage unidentifiable | 4 | 3 | ND | NA | NA | NA | NA |
| Total: | 310 | 298 | 29.5% (88/298) | 39.8% (35/88) | 11.4% (10/88) | 2.3% (2/88) | 46.6% (41/88) |

**Table 7. *Trypanosoma cruzi* molecular detection (n = 39/144) in correlation with bloodmeal hosts from Florida triatomine bugs.**

| Bloodmeal Host | *T. cruzi* TcI | *T. cruzi* TcIV | Mixed *T. cruzi* TcI & TcIV | *T. cruzi* untypable DTU |
|---|---|---|---|---|
| *Didelphis virginiana* (Virginia opossum) | 10.2% (4/39) | ND | ND | 20.5% (8/39) |
| *Anaxyrus terrestris/fowleri* (Southern or Fowler's toad) | 15.4% (6/39) | ND | ND | 5.2% (2/39) |
| *Dasypus novemcinctus* (Nine-banded armadillo) | 15.4% (6/39) | ND | ND | none |
| *Homo sapiens* (Human) | ND | ND | 2.6% (1/39) | 5.2% (2/39) |
| *Plestiodon laticeps* (Broad-headed skink) | ND | 2.6% (1/39) | ND | 5.2% (2/39) |
| *Hyla squirella* (Squirrel Tree Frog) | ND | 2.6% (1/39) | ND | 2.6% (1/39) |
| *Sylvilagus floridanus* (Eastern cottontail rabbit) | 2.6% (1/39) | ND | ND | ND |
| *Sciurus carolinensis* (Eastern gray squirrel) | 2.6% (1/39) | ND | ND | ND |
| *Canis lupus familiaris* (Domestic dog) | 2.6% (1/39) | ND | ND | ND |
| *Neotoma floridana* (Eastern woodrat) | ND | ND | ND | 2.6% (1/39) |
| *Felis catus* (House cat) | ND | ND | ND | 2.6% (1/39) |
| Total: | 48.7% (19/39) | 5.2% (2/39) | 2.6% (1/39) | 43.5% (17/39) |

ND – Not Detected; DTU – Discrete Typing Unit.

## Discussion

Our study confirms the wide geographic distribution of *T. sanguisuga* in Florida and supports historical records documenting its presence near human dwellings and its tendency to invade homes [15,16,42–44]. Further evidence of potential domiciliation (colonization of the human dwelling) is provided by the substantial prevalence of human bloodmeals. Nymphal stages of Florida *T. sanguisuga* were also commonly found near and some within human dwellings, which could indicate domiciliation. We detected *T. cruzi* among 29.5% of the triatomines tested (n = 88/298); both DTU TcI and TcIV were detected. Interestingly, among samples which were positive for *T. cruzi* genomic DNA, a large portion were unable to be typed using well validated methods (22,23,29). In the state of Texas, *T. sanguisuga* was more commonly found to harbor DTU TcIV rather than TcI [14], but among our specimens in Florida we found TcI was the dominant DTU. This is similar to *T. sanguisuga* analyzed in Louisiana which has found a predominance of TcI among their isolates [45] and may correlate with regional differences in *T. cruzi* ecological niches. Further work is needed to better understand the genomic diversity of *T. cruzi* strains and DTUs found in North America [46,47].

We found that *T. sanguisuga* fed on humans and Virginia opossums in peridomestic settings. The Virginia opossum is a wildlife reservoir for *T. cruzi* and can reach infection prevalences of nearly 50% in the peridomestic settings of Florida [22,23]. As in Central and South America, this finding would suggest that in Florida spillover of *T. cruzi* from its wildlife reservoirs into humans is possible and would further indicate that human exposure to *T. cruzi* in peridomestic settings is possible. The high proportion of triatomines with human blood found indoors (45.0%), relative to outdoors (12.5%) likely indicates that they enter homes readily to bite humans. This result would also indicate that some may move outdoors after feeding upon human blood or are feeding on humans in outdoor settings. Compromised window screens and loose-fitting doors or leaving unscreened doors and windows open are known entry points of triatomines. These perimeter points of entry may provide routes of movement in and out of a human dwelling [25,48].

The role of reptiles, amphibians and other non-mammalian hosts in the ecology of *T. cruzi* transmission needs additional study [45,49–52]. We found a large fraction (42.3%) of total bloodmeals from Florida *T. sanguisuga* collected outdoors fed upon amphibians or reptiles; in fact, just two ectotherm species, the southern toad and broad-headed skink, together accounted for 33.0% of total bloodmeals of triatomines captured outdoors. Findings of reptile and amphibian blood in triatomines have previously been documented, including in *T. sanguisuga* [45,52]. Reptiles, amphibians, and birds have largely been considered non-competent hosts/reservoirs for *T. cruzi*; however, some studies, including recent work, suggest that these animals may in fact be capable of being infected by *T. cruzi* [49,50,53]. Previous experimental work conducted in 1940 assessed transmission of *T. cruzi* by *T. sanguisuga* and revealed several important observations on the behaviors of this Florida triatomine [54]. Nymphs and adults collected from sabal palmetto leaf bases in Sarasota, Florida, fed on treefrogs, genus *Dryophytes* (formerly *Hyla*), which were abundant in the area. Intriguingly, Packchanian (1940) considered treefrogs to be important hosts for nymphal bugs in laboratory studies and field studies. Newly hatched bugs fed only on small frogs and tails of mice, but not on rats, guinea pigs, rabbits, or house sparrows, which he attributed to the thinner skin on frogs and mouse tails. Older bugs readily fed on these animals that nymphs were unable to bite successfully. Packchanian did not recover *T. sanguisuga* from "several hundred palmetto trees" that were examined for bugs in other areas of Florida (Jacksonville, Miami, Pensacola, Tallahassee, Tampa) or Georgia (Savannah) and attributed the absence of these triatomines to the absence of tree frogs in palm trees in those locations. These early field and experimental observations generally agree with our findings that Florida *T. sanguisuga* feed upon amphibians in nature. Several species of treefrog were fed upon (Table 4), including native (green treefrog, pine woods treefrog, and squirrel treefrog) and invasive (Cuban treefrog) species. While any bloodmeal host can help sustain a population of triatomines, whether reptiles and amphibians in Florida could serve to either amplify or reduce the *T. cruzi* infection prevalence in local triatomine populations is likely complex and should be further studied.

A few nymphal triatomines (n = 3) were found to have smoky brown cockroach hemolymph in their "bloodmeal", even though primers were not designed to amplify invertebrate DNA. *Triatoma sanguisuga* has been reported in sylvatic environments to cohabitate with cockroaches, such as the Florida wood roach (*Eurycotis floridana*), which can serve as source of food in times of starvation [20]. In a laboratory setting, *Triatoma recurva* (triatomine native to the southwestern U.S. and northern Mexico) was shown to survive solely on American cockroach hemolymph, although two other species (*Triatoma protracta* and *Triatoma rubida*) were not able to survive solely on this host [55]. Our investigation has shown a triatomine found in a natural setting with both cockroach hemolymph and human DNA within a bloodmeal. The significance of this finding is unclear, but given the relatively low percentage (2.5%) of cockroach DNA found in triatomine bloodmeals, this interaction is likely not a driving force in *T. sanguisuga* population abundance.

Florida *T. sanguisuga* has previously been described as its own subspecies known as *T. sanguisuga ambigua* [42–44]. Usinger (1944) described *T. s. ambigua* as the "common Florida subspecies" with slight color variations [43]. Frank Mead (1965) described *T. s. ambigua* specimens (n = 38) as ranging from the smallest being 16.1mm adult male collected in Gainesville, Florida, and the largest being an adult female measuring 20.4mm in Nassau County, Florida [44]. Lent and Wygodzinsky [15], agreed with Mead in that a "long series of specimens around the state is needed for a better understanding of the forms", which Lent and Wygodzinsky suggested may "apply to the entire *sanguisuga* complex" [15]. However, they rejected the subspecies classification of *T. sanguisuga* and to this date has not been adapted or further re-investigated. In our collection of Florida *T. sanguisuga* (N = 310) from 23 counties we did find varying color patterns and confirm our insects ranged within previously described sizes as described by Frank Mead (16–19mm). Further study is needed to understand whether Florida *T. sanguisuga* is indeed morphologically and genetically distinct from other *T. sanguisuga* insects found in other regions in the U.S. The identification of new *Triatoma* species has been recently described within the *Triatoma dimidiata* complex in Central America [56–58], supporting the notion that additional assessment may be warranted for *T. sanguisuga* in North America.

Our investigation assessed the presence of triatomines in Florida and the risk for *T. cruzi* transmission in the state. The evidence presented by this study suggests that all the necessary elements for vector-borne and possible oral transmission routes of Chagas disease are present. Triatomines were found in locations where humans would be at-risk for vector-borne and/or oral transmission routes (e.g., sleeping and food preparation areas). *Triatoma sanguisuga* has been shown to defecate after feeding which allows for stercoral transmission or depositing *T. cruzi*-laden feces to contaminate the surrounding environment [59,60]. As highlighted by a recent study, *T. sanguisuga* were found to have an average interval between feeding and first defecation at 9.75 minutes with 20% of these triatomines defecating within 2 minutes under laboratory conditions [60]. An evidence-based integrated pest management (IPM) strategy is in development for this vector and the findings of this investigation can contribute to this ongoing process [25,48]. Included in our IPM strategies is bringing public awareness to the public about triatomines and their potential to transmit Chagas disease in Florida [25]. Moreover, climate change and other anthropogenic landscape alterations in Florida and other regions of North America where *T. sanguisuga* is distributed may lead to more favorable conditions for geographical expansion of this vector of Chagas disease [61].

Some limitations of our investigation are noted. This assessment was centered on peridomestic collections, and we did not conduct extensive sylvatic or systematic explorations across all counties for triatomines. Our collection efforts were conducted in regions where triatomines had been reported from historical records or community scientists. We are also limited to data provided by community science participants regarding the collection of a triatomine. Collections in South Florida may have also been limited due to limited sampling efforts in these areas. Trapping protocols have not been established for triatomines in the United States, so our method of using UV lighting cannot be attributed to capture rate of triatomines, calling for further research in this area. Further systematic ecological data are needed to better understand the complete life cycle of triatomines in peridomestic and sylvatic habitats in Florida. Such data would support modeling for Florida *T. sanguisuga* natural environment throughout the state, which could help identify regions with high triatomine density and at-risk populations for potential autochthonous Chagas disease [61].

## Conclusions

*T. sanguisuga* are commonly found in peridomestic environments of human dwellings within the panhandle, north and central regions of Florida. Nymphal and adult stages of this triatomine were collected at homes, raising the possibility for domiciliation of the human dwelling. Invading triatomines harboring *T. cruzi* were collected by community scientists, and bloodmeal analysis demonstrated that they fed upon humans, canines, putative *T. cruzi* hosts (Virginia opossum), as well as reptiles and amphibians. Autochthonous Chagas disease has not been documented in Florida; however, conditions exist for autochthonous transmission of *T. cruzi* to humans and companion animals as demonstrated in our study. Further research is needed to understand these risks and the ecology of *T. cruzi* within sylvatic, peridomestic and domestic cycles in Florida [62].

## Supporting information

**S1 File.  Florida triatomine master list.**
(XLSX)

## Acknowledgments

We would like to thank our community scientists from Texas and Florida who participated in the program. We would also like to thank our lab volunteers from Florida: Bernardo Peniche, Sebastian Botero, Stephanie Katircioglu, Yasmin Tavares, Joshua Demetrius, Elizabeth Garcia, Madison Heisey and public community members for all their hard work and assistance in triatomine collections during our active manual searching in the field. We would also like to thank Bernardo Peniche for taking photos included in this paper.

## Author contributions

**Conceptualization:** Norman L. Beatty, Chanakya R. Bhosale, Rachel Curtis-Robles, Nathan Burkett-Cadena, Gabriel L. Hamer, Sarah A. Hamer, Samantha M. Wisely.

**Data curation:** Norman L. Beatty, Chanakya R. Bhosale, Zoe S. White, Carson W. Torhorst, Kristen N. Wilson, Rayann Dorleans, Tanise M.S. Stenn, Keswick C. Killets, Rachel Curtis-Robles.

**Formal analysis:** Norman L. Beatty, Chanakya R. Bhosale, Zoe S. White, Rachel Curtis-Robles, Nathan Burkett-Cadena, Gabriel L. Hamer, Sarah A. Hamer, Samantha M. Wisely.

**Funding acquisition:** Norman L. Beatty, Nathan Burkett-Cadena, Eva Nováková, Gabriel L. Hamer, Sarah A. Hamer, Samantha M. Wisely.

**Investigation:** Norman L. Beatty, Chanakya R. Bhosale, Carson W. Torhorst, Kristen N. Wilson, Tanise M.S. Stenn, Keswick C. Killets, Rachel Curtis-Robles, Eva Nováková, Sarah A. Hamer.

**Methodology:** Norman L. Beatty, Chanakya R. Bhosale, Zoe S. White, Kristen N. Wilson, Tanise M.S. Stenn, Keswick C. Killets, Rachel Curtis-Robles, Nathan Burkett-Cadena, Eva Nováková, Gabriel L. Hamer, Sarah A. Hamer, Samantha M. Wisely.

**Project administration:** Norman L. Beatty, Zoe S. White, Nathan Burkett-Cadena, Samantha M. Wisely.

**Resources:** Norman L. Beatty.

**Supervision:** Norman L. Beatty, Zoe S. White, Nathan Burkett-Cadena, Gabriel L. Hamer, Sarah A. Hamer, Samantha M. Wisely.

**Validation:** Norman L. Beatty, Chanakya R. Bhosale, Nathan Burkett-Cadena, Sarah A. Hamer, Samantha M. Wisely.

**Visualization:** Norman L. Beatty, Rachel Curtis-Robles.

**Writing – original draft:** Norman L. Beatty, Chanakya R. Bhosale, Nathan Burkett-Cadena.

**Writing – review & editing:** Norman L. Beatty, Chanakya R. Bhosale, Zoe S. White, Carson W. Torhorst, Kristen N. Wilson, Tanise M.S. Stenn, Keswick C. Killets, Rachel Curtis-Robles, Nathan Burkett-Cadena, Eva Nováková, Gabriel L. Hamer, Sarah A. Hamer, Samantha M. Wisely.

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
