## [Decision Letter · Decision Letter 0]

Response to Reviewers
Revised Manuscript with Track Changes
Manuscript

Shaden Kamhawi

co-Editor-in-Chief

Paul Brindley

co-Editor-in-Chief

**Journal Requirements:**

1) Please upload all main figures as separate Figure files in .tif or .eps format. For more information about how to convert and format your figure files please see our guidelines: 

2) Some material included in your submission may be copyrighted. According to PLOSu2019s copyright policy, authors who use figures or other material (e.g., graphics, clipart, maps) from another author or copyright holder must demonstrate or obtain permission to publish this material under the Creative Commons Attribution 4.0 International (CC BY 4.0) License used by PLOS journals. Please closely review the details of PLOSu2019s copyright requirements here: PLOS Licenses and Copyright. If you need to request permissions from a copyright holder, you may use PLOS's Copyright Content Permission form.

Potential Copyright Issues:

i) Please confirm (a) that you are the photographer of Striking Image, and Figures 1 & 3., or (b) provide written permission from the photographer to publish the photo(s) under our CC BY 4.0 license.

ii) Figure 5. Please confirm whether you drew the images / clip-art within the figure panels by hand. If you did not draw the images, please provide (a) a link to the source of the images or icons and their license / terms of use; or (b) written permission from the copyright holder to publish the images or icons under our CC BY 4.0 license. Alternatively, you may replace the images with open source alternatives. See these open source resources you may use to replace images / clip-art:

iii) Figure 2. Please (a) provide a direct link to the base layer of the map (i.e., the country or region border shape) and ensure this is also included in the figure legend; and (b) provide a link to the terms of use / license information for the base layer image or shapefile. We cannot publish proprietary or copyrighted maps (e.g. Google Maps, Mapquest) and the terms of use for your map base layer must be compatible with our CC BY 4.0 license.

3) We note that your Data Availability Statement is currently as follows: "All data made available.". Please confirm at this time whether or not your submission contains all raw data required to replicate the results of your study. Authors must share the “minimal data set” for their submission. PLOS defines the minimal data set to consist of the data required to replicate all study findings reported in the article, as well as related metadata and methods (https://journals.plos.org/plosone/s/data-availability#loc-minimal-data-set-definition).

- The points extracted from images for analysis..

4) Please ensure that the funders and grant numbers match between the Financial Disclosure field and the Funding Information tab in your submission form. Note that the funders must be provided in the same order in both places as well. State the initials, alongside each funding source, of each author to receive each grant. For example: "This work was supported by the National Institutes of Health (####### to AM; ###### to CJ) and the National Science Foundation (###### to AM).".

**Reviewers' comments:**

**Key Review Criteria Required for Acceptance?**

**Methods:**

-Are the objectives of the study clearly articulated with a clear testable hypothesis stated?

-Is the study design appropriate to address the stated objectives?

-Is the population clearly described and appropriate for the hypothesis being tested?

-Is the sample size sufficient to ensure adequate power to address the hypothesis being tested?

-Were correct statistical analysis used to support conclusions?

-Are there concerns about ethical or regulatory requirements being met?

Reviewer #1: Yes, the study effectively articulates its objectives. The study design is well-structured and appropriately tailored to address the stated objectives, utilizing a combination of community science and targeted fieldwork to maximize data collection. The triatomine collection methods and analytical tools are clearly described and relevant, ensuring meaningful ecological and epidemiological insights. The sample size is substantial, spanning a decade of data collection, which provides robust statistical power to support the findings. Additionally, the basic statistical and geo-mapping analyses are appropriately applied. This comprehensive approach enhances confidence in the study’s results and its implications for Chagas disease surveillance and transmission dynamics in the U.S.

Reviewer #2: Methods are sound. The only major comment is that a few methodological details are missing and as given, some of the methods used would not be repeatable by other researchers. In particular, the light trapping and wood roach DNA detection details are missing.

See attached PDF for line by line comments.

Reviewer #3: We know that community engagement is not easy to achieve. In this sense, it is important to describe what educational strategies were used by the project team in community engagement. Do local health services already conduct these educational actions? What is the flow of delivery of the triatomine by the resident to health services? The participation of community researchers needs to be better clarified.

Triatomines, unlike other hematophagous insects such as mosquitoes and phlebotomines, do not have an effective capture trap. The trap that classically has the highest capture rate is the Noireau trap. In this sense, it is noteworthy that the authors did not use it. Why was this trap not employed? It is important to include this clarification in the work. Furthermore, regarding the traps, the authors chose to use completely ineffective traps for capturing triatomines, such as ultraviolet light. Therefore, it is no surprise that no triatomines were captured using this method.

Reference:

Noireau F, Abad-Franch F, Valente SA, Dias-Lima A, Lopes CM, Cunha V, Valente V, Palomeque F, Carvalho-Pinto CJ, Sherlock I, Aguilar M, Steindel M, Grisard E, Jurberg J. Trapping triatominae in silvatic habitats. Memórias do Instituto Oswaldo Cruz 97: 61-63, 2002.

Between lines 163-164 - did these individuals undergo serological testing for T. cruzi? This is not clear in the text. I advise the authors to deposit representative vouchers of the species collected in the field in biological collections in the future. The importance and relevance of these deposits involving ecological studies can be appreciated in the articles below:

Thompson CW, Phelps KL, Allard MW, Cook JA, Dunnum JL, Ferguson AW, Gelang M, Khan FAA, Paul DL, Reeder DM, Simmons NB, Vanhove MPM, Webala PW, Weksler M, Kilpatrick CW. 2021. Preserve a voucher specimen! The critical need for integrating natural history collections in infectious disease studies. mBio 12:e02698-20. https://doi.org/10.1128/mBio.02698-20.

Salvador, R.B., Cunha, C.M. Natural history collections and the future legacy of ecological research. Oecologia 192, 641–646 (2020). https://doi.org/10.1007/s00442-020-04620-0

**Results:**

-Does the analysis presented match the analysis plan?

-Are the results clearly and completely presented?

-Are the figures (Tables, Images) of sufficient quality for clarity?

Reviewer #1: Yes, the analysis presented aligns well with the analysis plan, ensuring consistency between the study’s objectives and its methodological execution. The results are clearly and comprehensively presented, offering a thorough interpretation of the findings. The figures, including tables and images, are of high quality, effectively enhancing data clarity and accessibility. The study’s meticulous presentation strengthens its impact, making the findings both understandable and reproducible for future research in Chagas disease ecology and surveillance.

Reviewer #2: See attached PDF.

Reviewer #3: I suggest summarizing the text that repeats details contained within the tables (values, numbers, etc.).

Line 268: "However, this may be due to limited sampling efforts in these areas." This is discussion. Line 293: "triatomines, submitted by 31 individual contributors." How were these triatomines sent? This flow needs to be explained in the methods.

**Conclusions:**

-Are the conclusions supported by the data presented?

-Are the limitations of analysis clearly described?

-Do the authors discuss how these data can be helpful to advance our understanding of the topic under study?

-Is public health relevance addressed?

Reviewer #1: Yes, the conclusions are well-supported by the data presented, demonstrating a strong alignment between the findings and their interpretation. The authors clearly acknowledge the limitations of their analysis and pave the way for future much needed studies. Furthermore, they effectively discuss how these data contribute to advancing our understanding of Triatoma sanguisuga ecology, Trypanosoma cruzi transmission, and the broader implications for Chagas disease surveillance. Importantly, the study addresses the public health relevance of its findings, emphasizing the potential risk of autochthonous transmission and the need for increased awareness and surveillance efforts in the southeastern United States. As I mentioned in the general comments, this is a seminal study with a strong public health impact.

Reviewer #2: See attached PDF.

Reviewer #3: Has human serological testing been conducted in any region of the United States?

A limitation that needs to be added to the work: the group Triatominae does not have a good capture trap, as is the case with other insects, and the best on

**Editorial and Data Presentation Modifications?**

Reviewer #1: N/A

Reviewer #2: (No Response)

Reviewer #3: (No Response)

**Summary and General Comments:**

Reviewer #1: I have read with much interest the manuscript entitled: “Field Evidence of Trypanosoma cruzi Infection, Diverse Host Use and Invasion of Human Dwellings by the Chagas Disease Vector in Florida, USA, by Beatty and collaborators.

This study examines the distribution, host use, and Trypanosoma cruzi infection rates of Triatoma sanguisuga, a kissing bug species in Florida and the predominant in the US, using field sampling and community science from 2013 to 2023. Researchers collected 310 specimens from 23 counties, with over a third found inside homes and nearly 40% obtained through community participation. Molecular analysis revealed that 29.5% of tested bugs were infected with T. cruzi, primarily with Discrete Typing Units (DTUs) TcI (74.5%) and TcIV (21.3%). Bloodmeal analysis showed a diverse host range, including mammals (60%), ectothermic vertebrates (37%), and cockroaches (2.5%), with human blood detected in 23% of samples, highlighting significant human-vector contact. These findings importantly suggest that conditions in the southeastern U.S. may support local Chagas disease transmission and emphasize the need for further research on ectotherm involvement in T. sanguisuga and T. cruzi ecology.

This study by Dr. Beatty and collaborators represents a seminal contribution to the field, combining robust fieldwork with a decade-long community science initiative, echoing the national-scale efforts pioneered by the Hamer Lab at Texas A&M. The authors present an unprecedented, state-level landmark in the study of Triatoma sanguisuga ecology and Trypanosoma cruzi transmission dynamics, demonstrating a high level of reproducibility. Strengths include the extensive sampling period, detailed parasite DTU typing, and compelling ecological evidence that strongly suggests the likelihood of autochthonous Chagas disease transmission in the southern United States, particularly and for the first time in the State of Florida.

A remarkably impressive aspect is the inclusion of seasonality in Triatomine collections—an often-overlooked factor that has critical implications for disease transmission dynamics and surveillance programs. Beyond the characterization of T. cruzi DTUs, bloodmeal analysis, and the mapping of Triatomine distribution, this study sets a precedent for future state-level initiatives addressing this neglected parasitic disease.

My only recommendation to the authors is to clarify the delineation between specimens processed at Texas A&M and those later analyzed at the University of Florida/EPI. Having followed the remarkable work of this collaborative network, I understand the transition from Texas A&M’s initial contributions to Dr. Beatty’s establishment of independent laboratory capacity, but the general readership may not be aware. This study stands as an outstanding example of the power of research collaboration and paves the way for future investigations into Chagas disease ecology and surveillance in the U.S.

I strongly recommend acceptance of this manuscript.

Reviewer #2: In this important paper, the authors present a multitude of data demonstrating that Triatoma sanguisuga must be taken seriously as a Chagas disease vector in Florida and they highlight critical areas of further study. Line by line comments are attached in a PDF and pasted here:

Introduction

L70: vector’s *infected* excreta- indicate that the vector must be infected with T. cruzi, that it's not passed on by just the excreta

Methods

L144: what is Larry’s lighthouse- a vendor? A quick Google search does not return anything- please give details in case other researchers want to use the same methods

L145: specify ‘structures’

L145: does ‘walls around homes’ refer to the external walls of homes?

Fig 1B: Give details of UV light trapping for both the light sheets and the light traps (give details separately for each trap type): UV wavelength, when they were set up (i.e., time of day), for how many hours the traps were employed each night, #locations at which light trapping was carried out, how many nights the traps were used. etc. Include what worked and what didn’t, to the extent that other researchers can repeat (or not) the same methods.

L160: change ‘received’ to ‘sent’

167: it’s either ‘received by’ or ‘sent to’ but not ‘received to’

L167: morphologically identified how? Using which key?

L183: were negative controls included in the DNA extractions and the sequencing? How did you control for human contamination from the researchers handling the samples and doing the procedures?

L202 screened using which methods?

L202-203: fix grammar and repetitive ‘from’ in this fragment: ‘… utilizing similar methods from our group from previous studies among triatomines…’

L246-252: What BLAST e-value cutoff was used and why?

Results:

Table 4: insert subtotals for each taxon (eg amphibia, mammals, etc.)

General: How did they detect the wood roach DNA using vertebrate-specific primers. Please provide all methodological details to the extent that it is repeatable.

416: fix grammar

417-418: how was the toad associated with T. cruzi detection beyond the toads being a host for a single blood meal from six different infected bugs? We know triatomines take many blood meals in a lifetime, so it is a stretch to say that six infected bugs that fed on toads demonstrate that toads are associated with T. cruzi infection.

Table 5: Why did they calculate CIs here but not in any of the other proportional the results? The CIs are not discussed anywhere in the text and are more of a distraction than informative since they are not given with any context or explanation, and some of them are quite wide. Either cut them or add some interpretation as well as rationale for giving them for this metric but not any other ones.

Table 6: Break up last column into multiple columns, one for each type

Discussion

L524: change ‘defect’ to defecate

Reviewer #3: The work is comprehensive in addressing aspects related to the biological cycle of Chagas disease and has scientific merit. However, for publication, the authors need to add relevant information in the methodology section.

Introduction

In line 102, clarify whether you are referring to human or animal cases of Chagas. It is essential to provide a deeper ecological description of the species Triatoma sanguisuga: natural ecotope (animal burrows, tree hollows, association with oak trees?), association with which biomes? Preferred temperature and humidity, etc.

PLOS authors have the option to publish the peer review history of their article (what does this mean? ). If published, this will include your full peer review and any attached files.

**Do you want your identity to be public for this peer review?** For information about this choice, including consent withdrawal, please see our Privacy Policy .

Reviewer #1: No

Reviewer #2: No

Reviewer #3: No

**Figure resubmission:****Reproducibility:** To enhance the reproducibility of your results, we recommend that authors of applicable studies deposit laboratory protocols in protocols.io, where a protocol can be assigned its own identifier (DOI) such that it can be cited independently in the future. Additionally, PLOS ONE offers an option to publish peer-reviewed clinical study protocols. Read more information on sharing protocols at https://plos.org/protocols?utm_medium=editorial-email&utm_source=authorletters&utm_campaign=protocols

---

## [Decision Letter · Decision Letter 1]

Dear Beatty,

We are pleased to inform you that your manuscript 'Field Evidence of Trypanosoma cruzi Infection, Diverse Host Use and Invasion of Human Dwellings by the Chagas Disease Vector in Florida, USA' has been provisionally accepted for publication in PLOS Neglected Tropical Diseases.

Best regards,

Alessandra A Guarneri

Academic Editor

Nigel Beebe

Section Editor

Shaden Kamhawi

co-Editor-in-Chief

Paul Brindley

co-Editor-in-Chief

Reviewer's Responses to Questions

**Key Review Criteria Required for Acceptance?**

**Methods**

-Are the objectives of the study clearly articulated with a clear testable hypothesis stated?

-Is the study design appropriate to address the stated objectives?

-Is the population clearly described and appropriate for the hypothesis being tested?

-Is the sample size sufficient to ensure adequate power to address the hypothesis being tested?

-Were correct statistical analysis used to support conclusions?

-Are there concerns about ethical or regulatory requirements being met?

Reviewer #2: The authors have adequately addressed my initial comments.

Reviewer #3: (No Response)

**Results**

-Does the analysis presented match the analysis plan?

-Are the results clearly and completely presented?

-Are the figures (Tables, Images) of sufficient quality for clarity?

Reviewer #2: (No Response)

Reviewer #3: (No Response)

**Conclusions**

-Are the conclusions supported by the data presented?

-Are the limitations of analysis clearly described?

-Do the authors discuss how these data can be helpful to advance our understanding of the topic under study?

-Is public health relevance addressed?

Reviewer #2: (No Response)

Reviewer #3: (No Response)

**Editorial and Data Presentation Modifications?**

Reviewer #2: (No Response)

Reviewer #3: (No Response)

**Summary and General Comments**

Reviewer #2: (No Response)

Reviewer #3: (No Response)

PLOS authors have the option to publish the peer review history of their article (what does this mean? ). If published, this will include your full peer review and any attached files.

**Do you want your identity to be public for this peer review?** For information about this choice, including consent withdrawal, please see our Privacy Policy .

Reviewer #2: No

Reviewer #3: No

---

## [Editor Report · Acceptance letter]

Dear Dr. Beatty,

We are delighted to inform you that your manuscript, "Field Evidence of Trypanosoma cruzi Infection, Diverse Host Use and Invasion of Human Dwellings by the Chagas Disease Vector in Florida, USA," has been formally accepted for publication in PLOS Neglected Tropical Diseases.

Best regards,

Shaden Kamhawi

co-Editor-in-Chief

Paul Brindley

co-Editor-in-Chief
